# Daily feeding rhythm linked to microbiome composition in two zooplankton species

**Alaina Pfenning-Butterworth**[ID]◉*, **Reilly O. Cooper**◉, **Clayton E. Cressler**

School of Biological Sciences, University of Nebraska-Lincoln, Lincoln, Nebraska, United States of America

◉ These authors contributed equally to this work.
* apfenning@huskers.unl.edu

**Data Availability Statement:** Raw reads for the Daphnia magna and D. dentifera samples are available on NCBI (BioProject PRJNA715454). All code used to generate statistics and figures as well

## Abstract

Host-associated microbial communities are impacted by external and within-host factors, i.e., diet and feeding behavior. For organisms known to have a circadian rhythm in feeding behavior, microbiome composition is likely impacted by the different rates of microbe introduction and removal across a daily cycle, in addition to any diet-induced changes in microbial interactions. Here, we measured feeding behavior and used 16S rRNA sequencing to compare the microbial community across a diel cycle in two distantly related species of *Daphnia*, that differ in their life history traits, to assess how daily feeding patterns impact microbiome composition. We find that *Daphnia* species reared under similar laboratory conditions have significantly different microbial communities. Additionally, we reveal that *Daphnia* have daily differences in their microbial composition that correspond with feeding behavior, such that there is greater microbiome diversity at night during the host's active feeding phase. These results highlight that zooplankton microbiomes are relatively distinct and are likely influenced by host phylogeny.

## Introduction

The structure and function of host-associated microbial communities are linked with both intrinsic host factors and external factors. In particular, diet quality and variety influence microbiome composition; for example, mice transitioned from wild diets to controlled laboratory diets experienced shifts in microbiome composition and microbial functional pathways associated with carbohydrate metabolism, sugar metabolism, and motility [1]. Consequently, these compositional and functional changes correlate with changes in host behavior [2], host immune function [3], and other physiological functions [4].

Host diet is strongly influenced by feeding behavior, and many organisms show a daily rhythm in feeding behavior that is regulated by clock genes [5]. This rhythm could impact microbiome composition, both by altering diet-mediated microbial interactions and by modifying the intake of new microbes through feeding and the removal of existing microbes through peristalsis. However, there are few studies that link changes in feeding behavior and microbial composition, despite the downstream effects changes in the microbiome can have on host survival, fitness, and immunity [6].

To understand the links between biological rhythms, feeding behavior, and microbiome composition, we use the *Daphnia* model system. *Daphnia* are dominant herbivores and serve

as raw data are available on GitHub (https://github.com/reillyowencooper/circadian_microbiome).

**Funding:** The author(s) received no specific funding for this work.

**Competing interests:** The authors have declared that no competing interests exist.

as keystone species by linking primary producers with secondary consumers in freshwater systems [7]. *Daphnia* are an ideal system for addressing if and how feeding behavior and microbial composition are linked because (1) they have a relatively simple microbiome, which has been characterized in multiple species [8, 9], (2) they have core clock genes and show circadian expression of feeding behavior, immune, and sensory genes [10–13], and (3) quantifying individual feeding behavior is straightforward [14].

We examined two species of *Daphnia*, *D. dentifera* and *D. magna*. These species are not phylogenetically closely related, and they exhibit differences in life history traits [15, 16] that likely affect host-microbiome interactions. For instance, diel vertical migration (DVM), where *Daphnia* migrate to the top of the water column at night to feed [17], is thought to be an ancestral trait of all *Daphnia* species. However, more recent studies suggest that some species exhibit DVM more consistently than others. In *D. magna* DVM may be tightly linked to genotype [18, 19]; whereas *D. dentifera* show a robust DVM behavior in laboratory and natural systems regardless of genotype [20]. *Daphnia dentifera* also have a circadian rhythm in feeding behavior that is consistent with DVM [10]. Given these differences, we predicted that *D. dentifera*'s microbiome composition would be more diverse during their active phase (night). In contrast, we predicted that *D. magna* would not have differences in their microbiome composition across a day given that they are not known to exhibit a strong DVM.

We used 16S rRNA sequencing to compare the microbiome composition of individually reared *Daphnia* of each species to their feeding behavior across a diel cycle to address the following questions: (1) Are there microbiome composition differences among species of *Daphnia*? (2) Is there time of day differences in microbiome composition within each species and are the differences consistent with feeding behavior? (3) If there are daily differences in microbiome composition, are they consistent across both host species?

We found that *D. dentifera* and *D. magna* have significantly different microbiome composition. Additionally, microbiome composition does vary with time of day and is consistent with their feeding behavior, such that increased feeding and higher microbiome diversity occurred at night in both host species. However, there are host-specific differences in the abundance of specific microbes with time of day, suggesting that host feeding behavior and microbial community abundance are linked.

## Materials and methods

### Experimental design

Both *Daphnia dentifera* and *D. magna* were maintained in cultures for more than 2 and 5 years, respectively. Cultured individuals were maintained at 22˚C, in COMBO medium [21], under a 15:9 light:dark cycle and were fed batch-cultured *Ankistrodesmus falcatus* in 500 ml of high nitrogen COMBO under constant light at 22˚C. Individuals were moved to new COMBO and fed *A. falcatus* every other day at quantities relative to their body size (1 mgC/L for *D. dentifera* and 2.5 mgC/L for *D. magna*). At fourteen days old individuals were collected for DNA extraction.

### Feeding assay

Prior to the feeding assay, individual *D. dentifera* and *D. magna* were reared to 14 days old as described above with one exception. Individuals were fed 14 day old *A. falcatus* that was frozen in daily aliquots for the course of the feeding rate assay. This ensured that (1) individuals were fed a consistent chlorophyll:carbon ratio and (2) that algal quantity remained the same during feeding rate assays run in the dark versus in the light. Individual *D. dentifera* and *D. magna* were randomly assigned to either the day or night feeding rate assay (n = 14, N = 56).

To quantify feeding rate, we isolated *D. dentifera* and *D. magna* individuals in 15ml and 30 ml tubes (Nunc, Rochester, NY, USA) with 10 and 30 mL of COMBO media combined with 1mgC/L of *A. falcatus*, respectively. Individuals fed for six hours, after which they were moved to a new vial with fresh COMBO and their body size was measured. We measured algal fluorescence on a plate reader (Tecan, Maennedorf, Switzerland) following Hite et al. 2020, to compare the relative quantities of chlorophyll-*a* in wells with animals to control wells that did not contain animals [14]. We calculated individual feeding rates as:

$$\text{Feeding rate} = \ln\left(F_{\text{control}}/F_{Daphnia}\right) * v/t$$

where $F_{\text{control}}$ is the average of fluorescence of control wells, $F_{Daphnia}$ is the fluorescence of an animal well, v is the volume of COMBO and algae in ml, and t is the time individuals fed in hours [22]. We corrected each feeding rate by individual body size to account for species specific differences in feeding rate.

## DNA extraction and sequencing

*Daphnia magna* were pooled in sets of 5 individuals (n = 10 samples per time point) and *D. dentifera* in sets of 10 individuals (n = 10 samples per time point) for DNA extraction. DNA was extracted using the DNEasy Blood & Tissue Kit, with a 24h proteinase K digestion step to ensure that all microbial cells within and on hosts were digested. The 16S rRNA V4 hypervariable region was amplified using the 515f/806r primer pair [23] and PCR steps of 95˚C for 3 min; 35 cycles of 95˚C for 45 sec, 58˚C for 30 sec, 72˚C for 45 sec; and 72˚C for 5 minutes. Sample libraries were normalized with the SequalPrep Normalization Plate Kit, then quality checked using the KAPA Library Quantification Kit. Samples were pooled and PhiX spiked, then sequenced using the MiSeq Reagent Kit v2 (300 cycles) on the Illumina MiSeq (Nebraska Food for Health Center, Lincoln, NE, USA).

## Sequence analysis, visualization, and statistics

Demultiplexed samples were processed using the dada2 v1.16.0 [24] algorithm in R [25]. After read processing, amplicon sequence variants (ASVs) were assigned taxonomy using the dada2-formatted GTDB taxonomy reference database v3 [26]. Sample metadata, processed ASVs, taxonomic identity, and a phylogenetic tree constructed using the phangorn v2.5.5 [27] and DECIPHER v2.16.1 [28] packages were combined in a phyloseq v1.32.0 [29] object for downstream visualization and statistics. To analyze alpha diversity, the inverse Simpson index was estimated for each sample using DivNet v0.3.6 [30]. An ANOVA was used to analyze alpha diversity differences across host species and time points, then posthoc Tukey honest significant differences were calculated to understand differences among treatments. Beta diversity was measured across samples using weighted UniFrac distance, and statistically analyzed using PERMANOVAs across host species and time points. Core taxa in the microbiota of both species were identified using the microbiome package v1.10.0 [31], with taxa appearing across at least 50% of samples and at relative abundances ≥0.001 included in the analysis. To find taxa that differed significantly between time points, we focused on relatively common ASVs (≥0.005 relative abundance) and compared dayversus-night relative abundance using two statistical tests. For ASVs that were unique to one host species, we compared the relative abundance of each ASV at day versus night using a t-test. For ASVs abundant in both host species we compared the relative abundances of each ASV at day versus night using ANOVA.

## Results

### Composition of the microbiota is different across host species

Though both *Daphnia dentifera* and *D. magna* are freshwater zooplankton species and their rearing conditions are matched, their microbiomes differ substantially (Fig 1). We found that

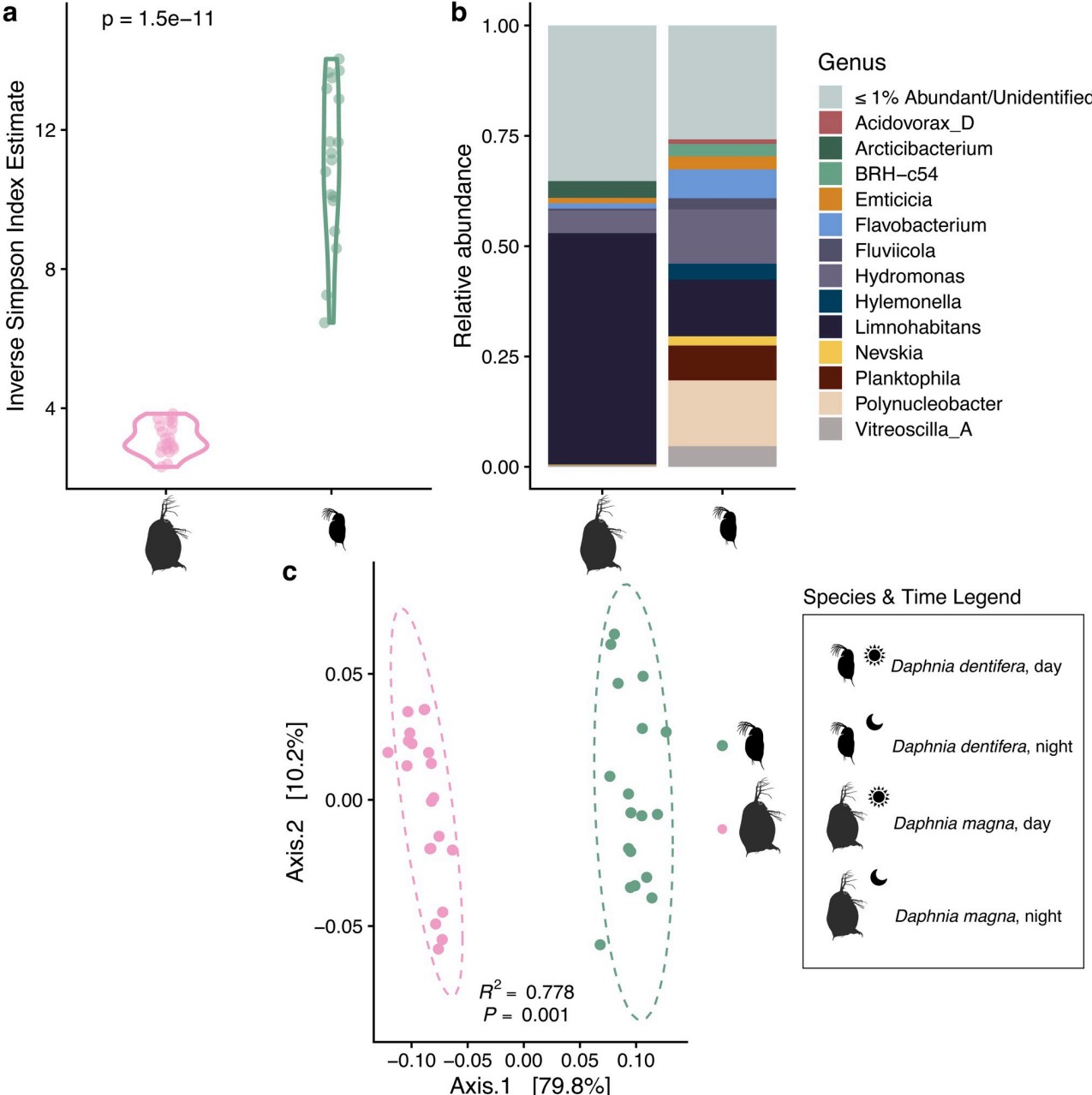

**Fig 1. Host species differences in microbiota structure.** a) Inverse Simpson index estimates for individual samples of *Daphnia magna* and *D. dentifera*, with t-test results reported. b) Genus-level bacterial microbiota composition for *D. magna* and *D. dentifera*. Amplicon sequence variants (ASVs) identified at $< = 1\%$ of relative abundance in each host species and ASVs not able to be identified at the Genus level are denoted by "$< = 1\%$ Abundant/Unidentified". c) Principal coordinate analysis using weighted UniFrac distances comparing *D. magna* and *D. dentifera*, with PERMANOVA results reported. Ellipses represent the multivariate t-distribution of *D. magna* and *D. dentifera* samples.

*D. dentifera* harbor a significantly more diverse set of bacterial taxa than *D. magna* (Fig 1a, Inverse Simpson index, *D. dentifera* = 11.39 ± 2.62, n = 19, *D. magna* = 3.13 ± 0.44, n = 20, $p < 0.0001$). The *D. dentifera* microbiome is not dominated by a single genus; instead, many genera exist at low abundances (Fig 1b). In contrast, the *D. magna* microbiome is dominated by the *Limnohabitans* genus (>50% relative abundance) and lacks many of the genera identified in the *D. dentifera* microbiome. In total, we find that community composition within each host species clusters together, indicating that *D. magna* and *D. dentifera* have statistically distinct bacterial communities (Fig 1c, PERMANOVA, pseudo-$F_{1, 37}$ = 124.47, $R^2$ = 0.778, $p = 0.001$).

## Composition of the microbiota and feeding behavior varies across a day

Between day and night, we observed changes in both feeding behavior and microbial community diversity and composition in both *Daphnia dentifera* and *D. magna* (Fig 2). Both host species had significantly more diverse microbiomes at night than during the day (Fig 2a, *D. magna*: t-test $p < 0.0001$, *D. dentifera*: t-test $p = 0.004$). In addition, systematic shifts in composition between day and night were observed in both species (Fig 2c). In *D. magna*, increased bacterial diversity at night corresponded with an increase in the relative abundance of *Arcticibacterium*, a decrease in *Limnohabitans*, and a large increase in the relative abundance of rare taxa each appearing at less than 5% abundance (increasing in total from a maximum relative abundance of 35% to a maximum of 55%), though no rare taxa became prominent enough to surpass any of the taxa consistently present at >5% relative abundance. *Daphnia dentifera* composition changes were more difficult to disentangle, but the more diverse night microbiome corresponded with lower abundances of *Hydromonas* and an increase in the relative abundances of rare taxa. We also found that both *Daphnia* species had significantly higher feeding rates during the night than the day (Fig 2b, *D. magna*: paired t-test $p = 0.008$, *D. dentifera*: paired t-test $p < 0.0001$).

## Specific amplicon sequence variants are associated with shifts in community composition across a day

To investigate the shifting microbiome composition in both host species beyond the genus taxonomic rank, we identified ASVs with relative abundances greater than .05% found only in *D. dentifera*, *D. magna*, or shared between the two species. We then examined whether relative abundance of these identified taxa differed during the day or night (Fig 3). Simultaneously, we identified ASVs that were core members of the microbiome in each species (S1 Table). We found seven ASVs that were considered core members in both host species, all belonging to the Gammaproteobacteria or Bacteroidia. We found a further 13 core ASVs in *D. magna* and 27 core ASVs in *D. dentifera*, though many of these ASVs were found at relative abundances less than 0.05%.

In ASVs specific to *D. magna*, only three experienced significant shifts between day and night: an *Arcticibacterium* ($p = 0.013$), *Emticicia* ASV 1 ($p = 0.038$), and a Polyangiaceae ($p = 0.004$) were more relatively abundant at night (Fig 3a). More unique ASVs were found in *D. dentifera*, and more were significantly different between time points: a *Polynucleobacter* ($p = 0.001$) and *Lacibacter* ($p = 0.047$) were more relatively abundant during the day, while *Limnohabitans* ASV 1 ($p = 0.017$), an NS11-12g ($p = 0.032$), and *Limnobacter* ($p = 0.001$) were more relatively abundant at night (Fig 3b). For ASVs found in both species (Fig 3c and 3d), only *Limnohabitans* ASV 3 was significantly different at the time points in both species. However, the abundance shift was opposite in the two species; in *D. magna*, *Limnohabitans* ASV 3 was more abundant during the day ($p < 0.0001$), but in *D. dentifera* it was more abundant at night ($p = 0.005$). In *D. magna*, a *Flavobacterium* ASV was more abundant at night ($p = 0.002$). The same *Flavobacterium* was more abundant during the night in *D. dentifera*, though not

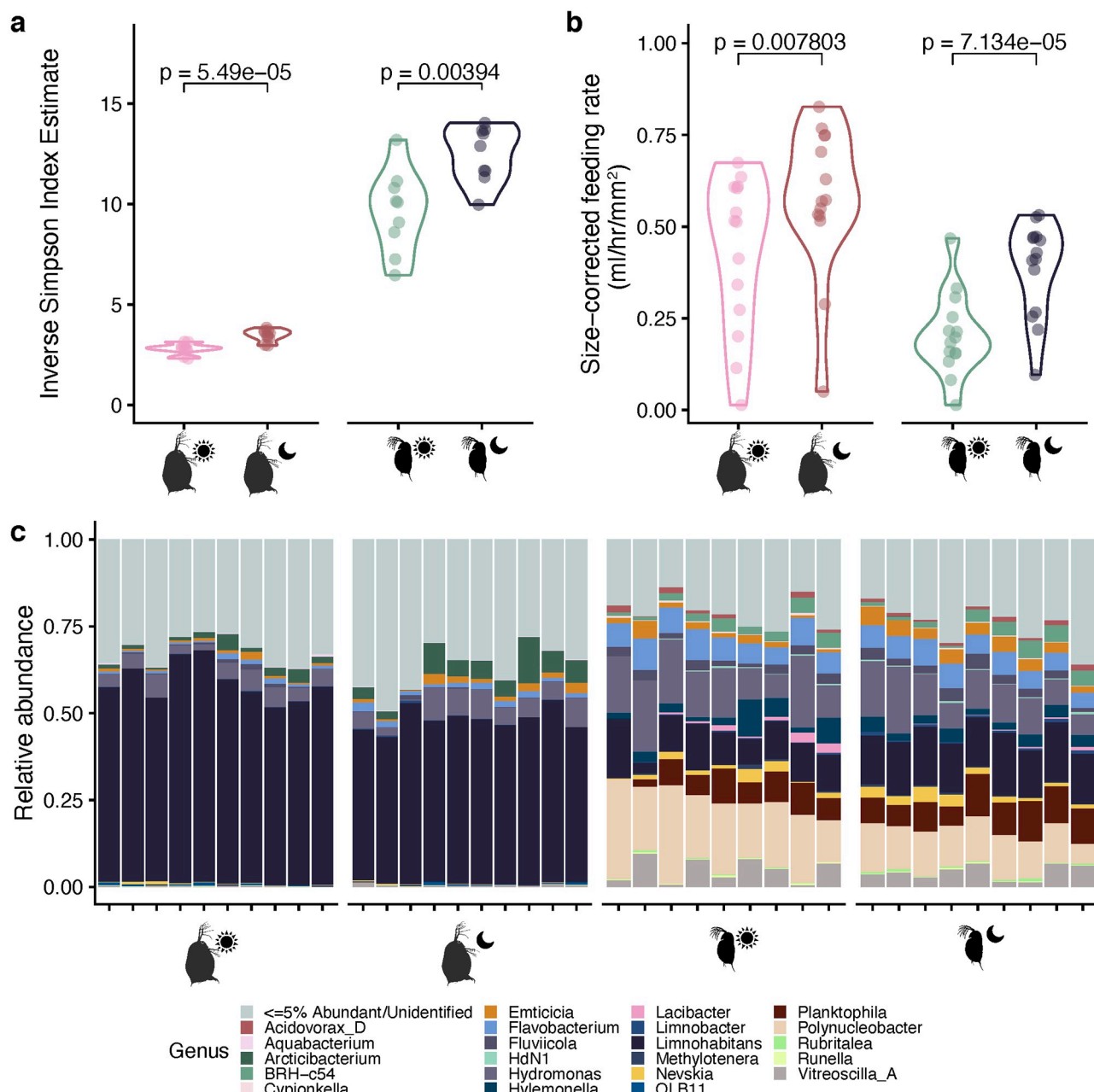

**Fig 2. Differences in microbiota structure at different times of day.** (a) Alpha diversity, represented by the inverse Simpson index, of *Daphnia magna* and *D. dentifera* during the day and night sample timepoints. For each species t-test results are reported. (b) Size-corrected feeding rates of *D. magna* and *D. dentifera* during the day and night sample time points (n = 14 per species). For each species t-test results are reported. (c) Genus-level composition plots for each sample taken for *D. magna* and *D. dentifera* at each timepoint (*D. magna*, n = 10; *D. dentifera*, n = 9).

significantly so (*p* = 0.06). In *D. dentifera*, a Chitinophagaceae ASV (*p* = 0.002) and *Emticicia* ASV 2 (*p* = 0.005) were more abundant at night.

## Discussion

Our results show that the microbiomes of *Daphnia dentifera* and *D. magna* are distinct. While most host-associated microbiomes are strongly associated with host phylogeny [32–36], the

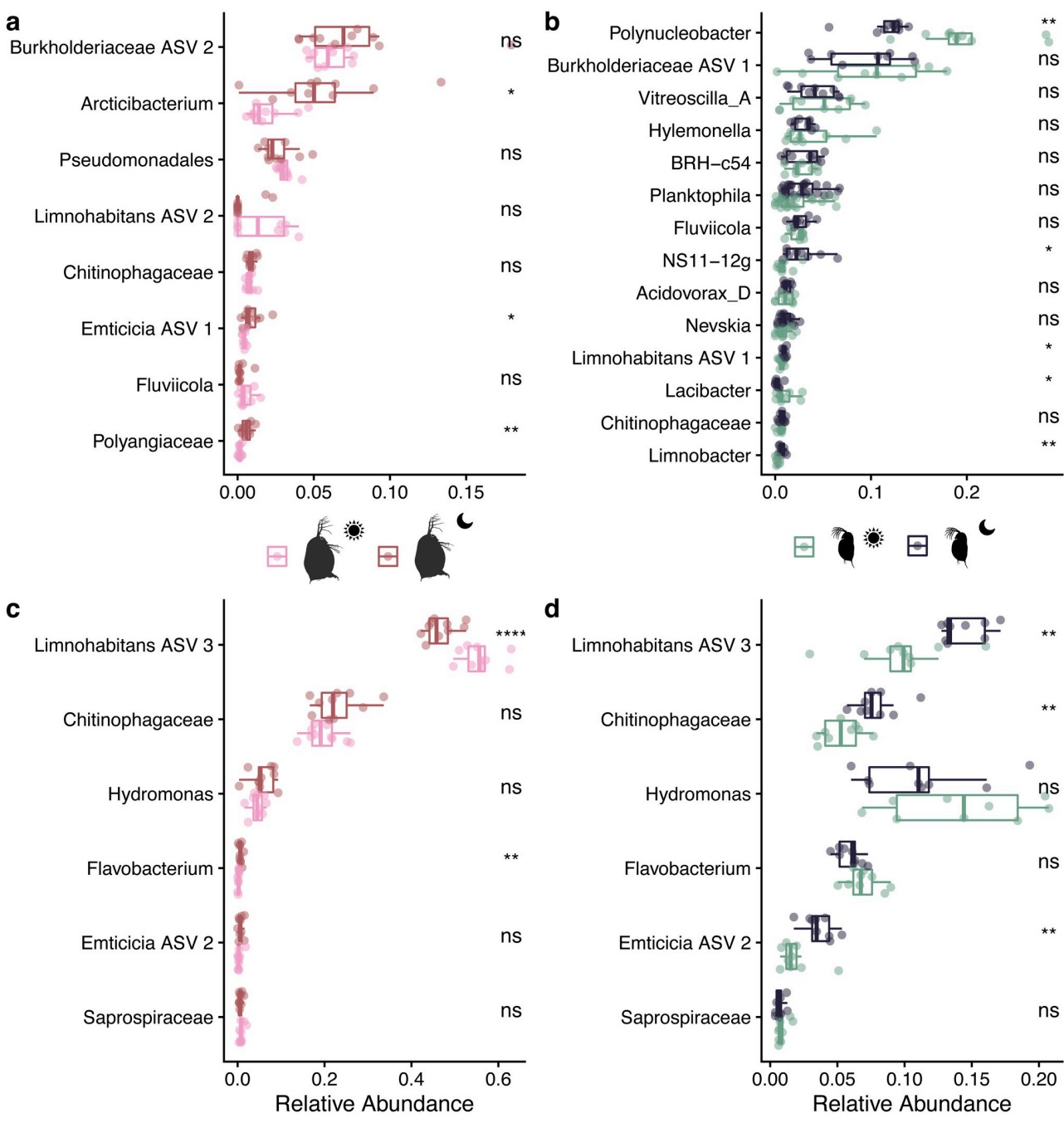

**Fig 3. Comparisons of ASVs with more than 0.5% abundance across all samples by host species and time point.** Significance of t-tests for each ASV are denoted on the right side of each figure (*, $p < 0.05$; **, $p < 0.01$; ***, $p < 0.001$; ****, $p < 0.0001$). (a) ASVs found only in *Daphnia magna* in day (pink) or night (red) samples. ASV taxonomic identity is denoted at the most specific taxonomic rank possible without assigning species-level identity. (b) ASVs found only in *D. dentifera* in day (green) or night (blue) samples, with taxonomic identity as in (a). (c) & (d) ASVs common to both host species. Relative abundances of ASVs in *D. magna* in (c) and relative abundances of ASVs in *D. dentifera* in (d), with taxonomic identity as in (a).

differences between these two species are striking, as these two distantly related species have been cultured in similar in-laboratory rearing conditions for years. While phylosymbiosis, the hypothesis that host-associated microbial communities of closely related host species are more similar to each other than to less related host species [37], has been observed in other

invertebrate species, this pattern is not frequently observed in zooplankton species, as much of their microbiome is acquired from the environment [8, 38]. The differences in community composition between *D. dentifera* and *D. magna* in this study, as well as the differences among other zooplankton species as demonstrated by Eckert et al. [8], provide some evidence that zooplankton microbiomes are relatively distinct. However, our results indicated that microbiome composition are not a result of environmental conditions.

Here, we find that *D. magna* has a far less diverse microbiome than *D. dentifera*. The microbiome of *D. magna* is consistently dominated by a *Limnohabitans* ASV and a Chitinophagaceae ASV, while the microbiome of *D. dentifera* is evenly distributed across several Burkholderiaceae ASVs (including *Limnohabitans*, *Hydromonas*, and other unidentified Burkholderiacae). Though both *Daphnia* species were reared in similar laboratory conditions their microbiomes were dramatically different. The *Daphnia dentifera* used here were originally collected from temperate lakes in Southern Michigan (USA) and the *D. magna* are from a pond at Kaimes Farm, Leitholm, Scottish Borders [39]. The microbiome composition differences observed may be remnants of the microbes in their natural habitat, despite having been reared in the lab for hundreds of generations. In concert with evidence that zooplankton have distinct microbiomes as compared to their local environment [9, 40, 41], these results suggest that these host species can actively shape and maintain their associated microbial communities.

Host behavior and the microbiome are intrinsically linked [42–44]. While the microbiomes of these zooplankton species are clearly distinct, the same patterns of increased bacterial diversity and increased feeding rate at night emerge here. This suggests that microbiome diversity is linked to feeding behavior in *D. dentifera* and *D. magna*, and likely both are tied to strong diel vertical migration behaviors present across *Daphnia* species [45–47]. Increased feeding rate at night may influence microbiome composition and diversity through dietary changes; across host species, diet is strongly associated with microbiome structure [4, 48–50]. Diet is also directly associated with host behavior, as both quality and quantity of available food influences how hosts forage [51–53]. Because diet can impact microbiome structure and composition, it is also likely that diet-derived nutrients taken up by taxa in the microbiome indirectly impact host behavior [54] and that there is a strong link between feeding behavior and the microbiome. Demonstrated microbiome-mediated behavioral changes include those involved in stress-related behavior [55] and social behavior [42, 56]. In *Daphnia* species, diel vertical migration (DVM) is a host behavior intrinsically tied to feeding. Zooplankton consume more food at night when migrating up the water column in the relative safety of darkness from visual predators, then return to darker, deeper layers of the water column in daylight, where they consume less food [10, 57]. Because *Daphnia* in this experiment were raised individually in 30 mL tubes, we demonstrate that changes in feeding behavior observed here are strongly associated with changes in microbial diversity in the absence of host migration through the water column.

Though it is difficult to infer the functional importance of taxa in the microbiota, prior work characterizing metagenome-assembled genomes of *Daphnia magna* has pinpointed some functions encoded by their microbes. In particular, two *Limnohabitans* species and one *Emticicia* species have had some functions characterized [58], which are likely the same species as the ASVs identified as *Limnohabitans* ASVs (*Limnohabitans* ASV 2 and 3, Fig 3) and the *Emticica* ASV (*Emticicia* ASV 2, Fig 3) in this study. Many of the functions encoded in these genomes involve amino acid biosynthesis and transport, vitamin and mineral transport, and carbohydrate degradation, and many pathways encoded by the whole *D. magna* metagenome are implicated in carbohydrate breakdown. This suggests that many of the taxa in the *Daphnia* microbiota are involved in sequestering usable nutrients from carbohydrates taken in while *Daphnia* feed.

In conclusion, our results suggest that zooplankton hosts may be able to maintain microbial communities associated with their natural environment even with substantial pressure of the current environment on zooplankton microbiome composition [8]. This work also emphasizes the link between daily rhythms in behavior, specifically feeding, and the microbiome. Incorporating metabolomics to infer functional differences could provide further insight into the host's metabolic processes. The continued pursuit to understand how and when daily rhythms in behavior and the microbiome impact within-host processes could provide key insights into metabolic and immune disorders.

## Supporting information

**S1 Table. ASVs that are core members of the microbiome in each host species.** The mean relative abundance in both host species is listed for each ASV as well as the taxonomic identity. (PDF)

## Acknowledgments

We thank Dr. Andrew Benson, Mallory Van Haute, and Qinnan Yang for their assistance with library preparation and amplicon sequencing. We credit Mathilde Cordellier for the *Daphnia magna* image used in figures.

## Author Contributions

**Conceptualization:** Alaina Pfenning-Butterworth, Reilly O. Cooper.

**Data curation:** Alaina Pfenning-Butterworth, Reilly O. Cooper.

**Formal analysis:** Alaina Pfenning-Butterworth, Reilly O. Cooper.

**Writing – original draft:** Alaina Pfenning-Butterworth, Reilly O. Cooper.

**Writing – review & editing:** Alaina Pfenning-Butterworth, Reilly O. Cooper, Clayton E. Cressler.

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
