## [Decision Letter · Decision Letter 0]

22 Dec 2021

PONE-D-21-36303Daily feeding rhythm linked to microbiome composition in two zooplankton speciesPLOS ONE

Dear Dr. Pfenning-Butterworth,

Thank you for submitting your manuscript to PLOS ONE. After careful consideration, we feel that it has merit but does not fully meet PLOS ONE’s publication criteria as it currently stands. Therefore, we invite you to submit a revised version of the manuscript that addresses the points raised during the review process.

We look forward to receiving your revised manuscript.

Kind regards,

Juan J Loor

Academic Editor

PLOS ONE

Journal Requirements:

2. Please include a copy of Table S1 which you refer to in your text on page 10.

Reviewers' comments:

Reviewer's Responses to Questions

**Comments to the Author**

1. Is the manuscript technically sound, and do the data support the conclusions?

Reviewer #1: Yes

2. Has the statistical analysis been performed appropriately and rigorously? 

Reviewer #1: Yes

3. Have the authors made all data underlying the findings in their manuscript fully available?

Reviewer #1: Yes

4. Is the manuscript presented in an intelligible fashion and written in standard English?

Reviewer #1: Yes

5. Review Comments to the Author

Reviewer #1: The original research presented by this group investigating the links between biological rhythms, feeding behaviors and microbial composition in Daphnia models was incredibly novel and interesting. While the data show that Daphnia species have daily differences in microbial composition that is directly related to feeding behavior, I have a few questions regarding some the statistical analyses.

In a Materials and Methods subsection titled "Sequence analysis, visualization and statistics", authors describe the use of individual t-test on amplicon sequence variants to examine taxa across time points. In many cases, the use of t-test on high-dimensional data can leave room for major error. If possible, can authors please answer the following question: Were there adjustments for p values and false discovery rates? Are there other studies that have used individual t-test to compare ASVs?

The Results and Discussion sections were well written and easy to follow. I appreciated the authors describing how the taxa affected during feeding phases may be involved in different metabolic processes. This also leaves room for deeper analysis and the development of future circadian rhythm-microbiome studies in other animal models.

Overall, I believe this manuscript could add to the overall scope of knowledge in this field.

6. PLOS authors have the option to publish the peer review history of their article (what does this mean?). If published, this will include your full peer review and any attached files.

Reviewer #1: No

---

## [Author Response · Author response to Decision Letter 0]

6 Jan 2022

Dear Editor, 

 With this I submit a revision of our manuscript, “Daily feeding rhythm linked to microbiome composition in two zooplankton species”, by myself, R.O. Copper, and C.E. Cressler. The original submission received a thoughtful review. We have revised the manuscript in accordance with the suggestion of the reviewer. Below we discuss our response (tabbed) to the reviewer’s comment.

In addition, we included a copy of S1 Table in the revision and updated citation 59 with the correct information. 

We hope you will find that we have addressed all the concerns of the reviewers satisfactorily, and that the revised manuscript will be suitable for publication.

Sincerely,

Alaina C. Pfenning-Butterworth

alainapfenning@gmail.com

Reviewer #1: The original research presented by this group investigating the links between biological rhythms, feeding behaviors and microbial composition in Daphnia models was incredibly novel and interesting. While the data show that Daphnia species have daily differences in microbial composition that is directly related to feeding behavior, I have a few questions regarding some the statistical analyses.

In a Materials and Methods subsection titled "Sequence analysis, visualization and statistics", authors describe the use of individual t-test on amplicon sequence variants to examine taxa across time points. In many cases, the use of t-test on high-dimensional data can leave room for major error. If possible, can authors please answer the following question: Were there adjustments for p values and false discovery rates? Are there other studies that have used individual t-test to compare ASVs?

 We used t-tests to compare the relative abundance of each individual ASV at day versus night within a host species (Lines 127-129). We did not compare the relative abundance of different ASVs to one another within the same host species. Because we are not conducting multiple comparisons within each host species-by-timepoint dataset and we are substantially reducing the high-dimensional dataset to only ASVs at relatively high abundance, we do not believe it is necessary to include a correction for multiple comparison. Other studies have used t-tests and ANOVAs to compare ASVs (see Wang et al. 2020 “Characterizing changes in soil microbiome abundance and diversity due to different cover crop techniques”; Lutz et al. 2019 “A Simple Microbiome in the European Common Cuttlefish, Sepia officinalis”; Aira et al. 2019 “Microbiome dynamics during cast ageing in the earthworm Aporrectodea caliginosa”). 

 We reworded the sentence to describe the comparison more clearly. Lines 127 – 132 now read: 

To find taxa that differed significantly between time points, we focused on relatively common ASVs (≥0.005 relative abundance) and compared day-versus-night relative abundance using two statistical tests. For ASVs that were unique to one host species, we compared the relative abundance of each ASV at day versus night using a t-test. For ASVs abundant in both host species we compared the relative abundances of each ASV at day versus night using ANOVA. 

The Results and Discussion sections were well written and easy to follow. I appreciated the authors describing how the taxa affected during feeding phases may be involved in different metabolic processes. This also leaves room for deeper analysis and the development of future circadian rhythm-microbiome studies in other animal models.

Overall, I believe this manuscript could add to the overall scope of knowledge in this field.

---

## [Decision Letter · Decision Letter 1]

21 Jan 2022

Daily feeding rhythm linked to microbiome composition in two zooplankton species

PONE-D-21-36303R1

Dear Dr. Pfenning-Butterworth,

We’re pleased to inform you that your manuscript has been judged scientifically suitable for publication and will be formally accepted for publication once it meets all outstanding technical requirements.

Kind regards,

Juan J Loor

Academic Editor

PLOS ONE

Additional Editor Comments (optional):

Reviewers' comments:

Reviewer's Responses to Questions

**Comments to the Author**

1. If the authors have adequately addressed your comments raised in a previous round of review and you feel that this manuscript is now acceptable for publication, you may indicate that here to bypass the “Comments to the Author” section, enter your conflict of interest statement in the “Confidential to Editor” section, and submit your "Accept" recommendation.

Reviewer #1: All comments have been addressed

2. Is the manuscript technically sound, and do the data support the conclusions?

Reviewer #1: Yes

3. Has the statistical analysis been performed appropriately and rigorously? 

Reviewer #1: Yes

4. Have the authors made all data underlying the findings in their manuscript fully available?

Reviewer #1: Yes

5. Is the manuscript presented in an intelligible fashion and written in standard English?

Reviewer #1: Yes

6. Review Comments to the Author

Reviewer #1: Special thanks to the authors for addressing the comments regarding statistics and rewording sentences for clarity. I believe the authors have addressed all of my comments and now the manuscript reads very well.

7. PLOS authors have the option to publish the peer review history of their article (what does this mean?). If published, this will include your full peer review and any attached files.

Reviewer #1: **Yes: **Meli'sa Shaunte Crawford

---

## [Editor Report · Acceptance letter]

24 Jan 2022

PONE-D-21-36303R1 

Daily feeding rhythm linked to microbiome composition in two zooplankton species 

Dear Dr. Pfenning-Butterworth:

I'm pleased to inform you that your manuscript has been deemed suitable for publication in PLOS ONE. Congratulations! Your manuscript is now with our production department. 

Kind regards, 

on behalf of

Dr. Juan J Loor 

Academic Editor

PLOS ONE